

# Simultaneous shipborne measurements of $CO_2$, $CH_4$ and CO and their application to improving greenhouse gas flux estimates in Australia

Beata Bukosa[1], Nicholas M. Deutscher[1], Jenny A. Fisher[1,2], Dagmar Kubistin[1,3], Clare Paton-Walsh[1], and David W. T. Griffith[1]

[1]Centre for Atmospheric Chemistry, School of Chemistry, University of Wollongong, NSW, Australia
[2]School of Earth and Environmental Sciences, University of Wollongong, NSW, Australia
[3]German Meteorological Service, Meteorological Observatory Hohenpeissenberg, Hohenpeissenberg, Germany

*Correspondence to:* Beata Bukosa (bb907@uowmail.edu.au)

**Abstract.** Quantitative understanding of the sources and sinks of greenhouse gases is essential for predicting greenhouse gas-climate feedback processes and their impacts on climate variability and change. Australia plays a significant role in driving variability in global carbon cycling, but the budgets of carbon gases in Australia remain highly uncertain. Here, shipborne Fourier Transform Infrared Spectrometer measurements collected around Australia are used together with a global chemical
transport model (GEOS-Chem) to identify and quantify the sources of three direct and indirect carbon greenhouse gases: carbon dioxide ($CO_2$), methane ($CH_4$) and carbon monoxide (CO). Using these measurements, we provide an updated distribution of these gases and their sources and sinks. We find that for all three gases, the east Australian coast is largely influenced by local anthropogenic sources, which can be transported as far as 400km off the coast. The south and west coasts are characterised by a mixture of anthropogenic sources and biomass burning. Tropical northern regions are dominated by biomass burning emissions,
with significant contribution from fossil fuel for $CO_2$ and wetlands for $CH_4$. Averaged across Australia, fossil fuels followed by biomass burning contribute the most to total $CO_2$ and to both its background value and short-term enhancements. Wetlands provide the largest background $CH_4$ source, followed by livestock, oil, gas and waste emissions, with short-term enhancements mainly driven by anthropogenic sources. For CO, secondary production from oxidation of $CH_4$ and non-methane volatile organic compounds contributes most to the background and total CO burdens, while enhancements are driven by biomass
burning and anthropogenic sources. Clean air characteristic of the tropospheric background was observed away from the coast in the Indian Ocean, Coral Sea, and Tasman Sea. From the measurements in the Indian Ocean, we found that the background values of all three gases increase towards the tropics with latitudinal gradients of 0.019±0.003 ppm deg$^{-1}$ for $CO_2$, 0.34±0.02 ppb deg$^{-1}$ for $CH_4$ and 0.82±0.05 ppb deg$^{-1}$ for CO. Comparing coincident and co-located enhancements in the three carbon gases highlighted several common sources from the Australian continent. We found evidence for 17 events with similar
enhancement patterns indicative of co-emission and calculated enhancements ratios and modelled source contributions for each event. We found that anthropogenic co-enhancement events are common along the east coast, while co-enhancement events in the tropics primarily derive from biomass burning sources. Few co-enhancement events were observed along the south and west coasts. While the GEOS-Chem model generally reproduced the timing of co-enhancement events, it was less able to





reproduce the magnitude of enhancements. We found model overestimates of $CH_4$ from coal burning and underestimates of all three gases from biomass burning with overestimates for CO during some events. We identified missing sources from fossil fuel, biofuel, oil, gas, coal, livestock, biomass burning and the biosphere in the model, pointing to the need to further develop and evaluate greenhouse gas emission inventories for the Australian continent.

## 1   Introduction

Carbon greenhouse gas emissions to the atmosphere have grown dramatically over the last 250 years, with resulting impacts for climate. Before the industrial revolution, these gases were primarily controlled by natural processes, but since industrialization anthropogenic processes have played an increasingly important role in determining greenhouse gas budgets. This change has increased the complexity of the greenhouse gas-climate feedback and the uncertainties related to these feedbacks and processes. Carbon dioxide ($CO_2$) and methane ($CH_4$) are the most significant greenhouse gases arising from anthropogenic activities. Carbon monoxide (CO) is an indirect greenhouse gas that, through its reaction with the hydroxyl radical (OH), affects the atmospheric burdens of $CH_4$ and tropospheric ozone. The Australian continent has been shown to critically influence the interannual variability of carbon cycling on a global scale (Poulter et al., 2014), yet the budgets of these gases in Australia remain poorly constrained. Here, we use shipborne observations of $CO_2$, $CH_4$, and CO to provide an updated estimate of their spatial distribution, sources and sinks, with a focus on common processes and sources that lead to co-variation between species.

Australia plays a significant role in global carbon cycling. Recent studies showed that Australia, as a semi-arid region, was responsible for more than half of the 2011 global carbon sink anomaly and can be a dominant driver of the interannual variability in the global carbon cycle (Poulter et al., 2014; Ma et al., 2016) due to asymmetry in the interannual distribution of rainfall and in the response of the ecosystem to rainfall change (Haverd et al., 2017). Strong La Niña-induced wet periods lead to exceptionally large vegetation productivity, with associated large carbon sink events expected to be observed in the future. However, the vulnerability of this episodic land carbon sink to drought can lead to a rapid reversal of this $CO_2$ uptake, releasing the sequestered carbon back to the atmosphere (Ma et al., 2016).

Constraints on the processes controlling carbon gas variability and trends in Australia are therefore pivotal for accurately simulating global greenhouse gas variability and impacts. There have been several prior attempts to identify source contributions to Australian greenhouse gas budgets. The terrestrial biosphere is thought to be the largest driver of both column and surface $CO_2$ variability in Australia, followed by biomass burning (Deutscher et al., 2014; Buchholz et al., 2016). For $CH_4$, emissions from ruminant animals are a significant Australian source, particularly at clean air sites (Dalal et al., 2008; Fraser et al., 2011). Local emissions from animals are also present in urban areas, along with coal mining, biomass burning and wetland emissions. Wetlands are particularly important in the tropics where their emissions dominate (Deutscher et al., 2010). For CO, biomass burning plays an important role as the main driver of the CO seasonal and interannual variability across the southern hemisphere (Edwards et al., 2006a, b). Overall, total CO in Australia is dominated by non-methane volatile organic carbon (NMVOC) and $CH_4$ oxidation (Té et al., 2016; Fisher et al., 2017), with negligible influence from anthropogenic emis-



sions (Zeng et al., 2015). While prior work has provided some constraints on Australia's greenhouse gas sources, both these studies and others have shown lingering differences between modelled and measured concentrations, implying some sources of greenhouse gases in Australia remain missing or underestimated (Fraser et al., 2011; Loh et al., 2015).

Long range transport and interhemispheric exchange additionally influence the abundances of $CO_2$, $CH_4$ and CO in Aus-
tralia and confound measurement interpretation. The Intertropical Convergence Zone (ITCZ) and chemical equator (Hamilton et al., 2008) serve as a barrier to mixing between the more polluted northern hemisphere and cleaner southern hemisphere air (Stehr et al., 2002). During austral summer the ITCZ stretches across northern Australia, which chemically becomes part of the northern hemisphere, and during the austral monsoon season the chemical equator separates from the ITCZ north of Australia. South of the South Pacific Convergence Zone (part of the ITCZ) emissions from biomass burning are readily transported to
northern Australia from Indonesia and Southeast Asia (Gregory et al., 1999; Paton-Walsh et al., 2010; Fraser et al., 2011; Yashiro et al., 2009). Southeast Australia is also affected by long-range transport of biomass burning emissions, largely from southern Africa and South America (Jones et al., 2001; Edwards et al., 2006a; Zeng et al., 2012).

While most prior work on greenhouse gas source attribution in Australia has focused on a single species, measurements of co-variation between species can provide useful constraints on controlling processes (Andreae and Merlet, 2001; Popa et al., 2014).
$CO_2$, $CH_4$ and CO are chemically dependent, with several common sources and sinks, and changes in any one of these species can have a significant impact on the others. Both CO and $CH_4$ are removed through reaction with OH, the main tropospheric oxidant, leading to production of $CO_2$ (McConnell et al., 1971; Hewitt and Harrison, 1985; Enting and Mansbridge, 1991; Duncan et al., 2007). $CH_4$ oxidation leads to a near unity production of CO (Duncan et al., 2007), and CO oxidation is responsible for about 90% of the chemical production of $CO_2$ (Ciais et al., 2008; Folberth et al., 2005). All three gases
are emitted during fossil fuel and biomass combustion. Because of these co-emissions that lead to coincident enhancements, ratios between the different gases can be used to identify the signature of sources including coal mining (Buchholz et al., 2016), household combustions (Zhang et al., 2000), traffic (Ammoura et al., 2014), and biomass burning (Nara et al., 2011; Parker et al., 2016). Nonetheless, few studies have exploited the benefits of multi-species analysis to explore co-variations and constrain relevant source and sink processes of $CO_2$, $CH_4$ and CO in Australia.
In this study, we use 6 months of observations from 2012-2013 collected onboard a ship that circumnavigated Australia (Sect. 2), combined with a chemical transport model (GEOS-Chem, Sect. 3), to quantify the distributions of $CO_2$, $CH_4$ and CO around Australia (Sect. 4). We investigate the role of different sources and sinks in driving the variability of these gases (Sect. 5) by identifying a series of events when we observed simultaneous enhancements in at least two of the three gases. Finally, we use these enhancements and their co-variations to identify the dominant processes driving carbon gas variability in
Australia and to identify the sources that remain missing or underestimated in the GEOS-Chem model (Sect. 6).

## 2 Measurements

$CO_2$, $CH_4$ and CO were measured aboard the Australian research vessel Southern Surveyor operated by CSIRO/MNF (Commonwealth Scientific and Industrial Research Organisation/Marine National Facility) during seven voyages in austral autumn,



**Table 1.** FTIR analyser 5 min repeatability and accuracy for $CO_2$, $CH_4$ and CO.

| Trip | Repeatability | Accuracy |
|------|--------------|----------|
| $CO_2$ (ppm) | 0.06 | 0.15 |
| $CH_4$ (ppb) | 0.6 | 0.7 |
| CO (ppb) | 0.7 | 0.7 |

winter and spring 2012 and 2013 (Supplement, Table S1). Figure 1 shows the locations of the ship measurements. In 2012 the voyage started in Hobart (April), after which the ship went northeast to Brisbane (Trip 1, May) then turned towards Fiji (Trip 2, May) and returned to Hobart (Trip 3, June). The 2013 trip also started from Hobart (June), after which the ship turned west towards Perth (Trip 4, June) and proceeded clockwise to Broome (Trip 5, July) and along northern Australia (August) then south to Brisbane (Trip 6, September) and back to Hobart (Trip 7, October). For the analysis we separated the data into northbound (NB) and southbound (SB) sections for both years (Figure 1).

The measurements and data analysis are described in detail in a forthcoming paper in Earth System Science Data (Kubistin et al.) and are briefly summarised here. The data will be available in Pangaea. All trace gas mole fractions were measured with a Fourier Transform Infrared (FTIR) trace gas analyser which was an early version of that described by Griffith et al. (2012) (see also Esler et al. (2000)). The analyser is based around a Bruker IRcube FTIR spectrometer coupled to a 22m multipass White cell containing the sampled air. Trace gas amounts are retrieved from the collected spectra by least squares fitting of calculated spectra to the measured spectra in four spectral regions between 2000 and 3800 $\mathrm{cm}^{-1}$ (Griffith, 1996; Griffith et al., 2012). Sampled air from the foremast of the ship flowed at 1 $\mathrm{Lmin}^{-1}$ through the measurement cell. Single 1 s spectra were measured continuously and averaged over 5 min for the 2012 and 3 min for the 2013 voyage. The analyser was calibrated before and after the voyages against a suite of standard reference gases provided by CSIRO with assigned mole fractions on the relevant World Meteorological Organization - Global Atmosphere Watch (WMO-GAW) scales - WMO X2007 scale for $CO_2$, X2004A for $CH_4$ and X2014 for CO. During the voyages the calibration was checked against a single calibrated target tank and adjusted as required.

Precision and accuracy were determined from 5 min Allan Variance and 1 sigma reproducibility of the target tank measurements respectively. Table 1 summarises the 5 min repeatability and accuracy for each species.

## 3 Model description

To investigate the sources and sinks driving the measured carbon greenhouse gases, we used the GEOS-Chem 3D global chemical transport model (Bey et al., 2001). The meteorological inputs for GEOS-Chem come from the Modern-Era Retrospective analysis for Research and Applications, Version 2 (MERRA2) reanalysis developed by the NASA Global Modelling and Assimilation Office (GMAO). We use the offline $CO_2$, $CH_4$ and CO simulations from GEOS-Chem v11-01. The $CO_2$ simulation





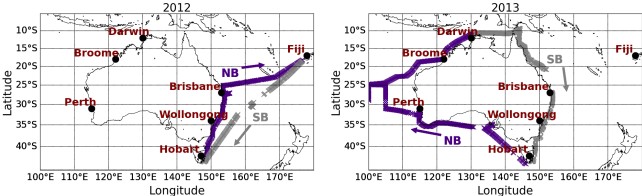

**Figure 1.** Locations of the shipborne measurements (purple/grey) and other sites relevant for the data interpretation (red). The ship track is separated into northbound (NB - purple, 147.5-176.6°E in 2012 and 146.1-130.9°E in 2013) and southbound (SB - grey, 176.6-146.1°E in 2012 and 130.9-147.5°E in 2013) sections to ease the interpretation of the data.

is based on Nassar et al. (2010) and Nassar et al. (2013), the $CH_4$ simulation is based on Wecht et al. (2014), and the CO simulation is described by Fisher et al. (2017).

We ran the model at 2°x2.5° horizontal resolution with 47 vertical levels from January 2005 through December 2014. The simulations were initialized with a 10-year spinup for $CO_2$ and $CH_4$ using 2005 as a base spinup year and a 6-month spinup for

CO using 2005. We have found these spinup periods to be sufficient to establish consistent spatial gradients in the atmosphere of all the tracers and total amount of each gas. The emission inventories and chemical fields used by each simulation are shown in Table **??**. Where possible, we used common emission inventories for all three simulations. The three carbon gas simulations are decoupled, hence the chemical production and loss of each species (e.g., $CO_2$ production from the oxidation of CO, $CH_4$ and NMVOCs) were computed offline using archived production rates and OH concentrations. For simulations that

were outside of the specified inventory time range, the model re-used the data from the closest year. The lack of time specific emission inventories can introduce uncertainties in the results, but we expect these errors to be low in our simulations (Fisher et al., 2017; Nassar et al., 2010).

The carbon gas simulations are all linear, and for each we included a suite of tracers tagged by source type (and, for CO, region). The tagged $CO_2$ simulation includes 9 tracers to distinguish between source types: fossil fuel, ocean exchange, biomass

burning, biofuel, balanced biosphere, net annual terrestrial exchange, shipping, aviation, and the collective $CO_2$ production from the oxidation of CO, $CH_4$ and NMVOCs. The ocean exchange, balanced biosphere, and net annual terrestrial exchange act both as a sink and source, while the other tracers represent only sources of $CO_2$.

The $CH_4$ tagged simulation includes 11 tracers for different source types: gas and oil, coal, livestock, waste, biofuel, rice cultivation, biomass burning, wetlands, termites, soil absorption and other combined anthropogenic emissions (e.g., energy

manufacturing transformation, non-road transportation, road transportation, industrial process and product use, and fossil fuel fires). The soil absorption represents a sink of $CH_4$ while all other tracers are sources. For $CH_4$, an OH sink is applied to all of the tracers; however, in contrast to the soil absorption sink there is no separate tracer for this loss.

The CO tagged simulation includes 4 source types: anthropogenic, biomass burning, and separate $CH_4$ oxidation and NMVOC oxidation. The anthropogenic tracer includes both fossil fuel and biofuel since these sources are combined in some

of the emission inventories. Stratospheric and tropospheric OH sinks are applied to all of the CO tracers. We further dis-





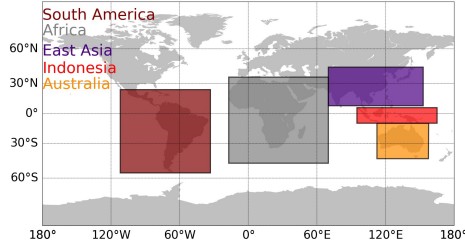

**Figure 2.** GEOS-Chem tagged regions used for anthropogenic and biomass burning sources in the CO simulation. South America ($112^\circ$W - $33^\circ$W, $56^\circ$S - $24^\circ$N), Africa ($17^\circ$W - $70^\circ$E, $48^\circ$S - $36^\circ$N), East Asia ($8^\circ$N - $45^\circ$N, $70^\circ$E - $153^\circ$E), Indonesia ($10^\circ$S - $6^\circ$N, $95^\circ$E - $165^\circ$E), Australia ($44^\circ$S - $10^\circ$S, $112.5^\circ$E - $157.7^\circ$E). The East Asia and Indonesia regions were used for the biomass burning source only. Regions not shown on the map are Anthropogenic Other and Biomass Burning Other, these are regional tags that cover everything except the source specific tagged regions.

tinguished the anthropogenic and biomass burning tracers by region to aid in interpretation of transported influences. The transported amounts of the anthropogenic and biomass burning sources hereinafter refer to emissions from the non-Australian tagged regions as shown in Figure 2.

For comparison to the ship measurements, model outputs were saved for grid boxes corresponding to the measured time,
latitude and longitude along the ship track at the model surface level. Both the measurements and modelled output were averaged to the model temporal (20 min) and spatial ($2^\circ$x$2.5^\circ$) resolution to calculate one average value for each unique grid box-timestep combination. Hereinafter we will refer to this averaging method as the measurement-model averaging.

The model initial conditions and the imbalance between the modelled sources and sinks relative to the their true values created a bias in the model, which led to a difference between the modelled and measured growth rates. To compare our surface
$CO_2$ and $CH_4$ measurements with the model, we corrected the modelled growth rates by first assessing offsets between the modelled and measured surface values at background stations (Barrow, Trinidad, Mauna Loa, American Samoa, Cape Grim and South Pole) (Dlugokencky et al., 2018b, a) as shown in Figure S1 in the Supplement. The modelled offset was then corrected with a globally-averaged 13-point running mean of the difference between the modelled and measured data at the background sites. We applied this linear correction method for $CO_2$ and $CH_4$. CO was not affected by this bias due to its shorter lifetime
and lack of long term trend.





**Table 2.** GEOS-Chem emission inventories and chemical fields used for the three carbon gas simulations. Years represent periods when time-specific inventories were available during our simulation time period (2005-2014).

| Source and sink fields | $CO_2$ | Years | $CH_4$ | Years | CO | Years |
|---|---|---|---|---|---|---|
| Anthropogenic[a] | ODIAC[b] | 2005-2014 | EDGARv4.2[c] | 2005-2008 | EDGARv4.2[c] | 2005-2008 |
| *Europe* | - | - | - | - | EMEP[d] | 2005-2012 |
| *Mexico* | - | - | - | - | BRAVO[e] | - |
| *Canada* | - | - | - | - | CAC[f] | 2005-2008 |
| *USA* | - | - | - | - | NEI[g] | 2006-2013 |
| *Asia* | - | - | - | - | MIX v1.1[h] | 2008-2010 |
| Biomass Burning | QFEDv2[i] | 2005-2014 | QFEDv2[i] | 2005-2014 | QFEDv2[i] | 2005-2014 |
| Biofuel Burning | Yevich and Logan[j] | - | - | - | Yevich and Logan[j] | - |
| Ocean Exchange | Takahashi et al.[k] | 2000-2013 | - | - | - | - |
| Balanced Biosphere | SIB3[l] | 2005-2010 | - | - | - | - |
| Net Terrestrial Exchange | TransCom[m] | - | - | - | - | - |
| Shipping | ICOADS[n] | - | - | - | ICOADS[n] | - |
| Aviation | AEIC[o] | 2005 | - | - | AEIC[o] | 2005 |
| Soil and Termites | - | - | Fung et al.[p] | - | - | - |
| Wetland | - | - | WetCHARTs v1.0[q] | 2005-2014 | - | - |
| Rice | - | - | EDGARv4.3.2[c] | 2009 | - | - |
| $P(CO)_{CH_4}$[r] | - | - | - | - | Archived fields[s] | 2005-2011 |
| $P(CO)_{NMVOC}$[r] | - | - | - | - | Archived fields[s] | 2005-2011 |
| $P(CO_2)$[t] | Archived fields[u] | 2005-2009 | - | - | - | - |
| OH sink | - | - | Archived fields[v] | - | Archived fields[s] | 2005-2011 |

[a] The anthropogenic emissions in the CO simulation had regional overwrites for the countries specified in the table.
[b] Open-source Data Inventory of Anthropogenic $CO_2$ (Oda and Maksyutov, 2011)
[c] European Commission. Emission Database for Global Atmospheric Research (http://edgar.jrc.ec.europa.eu/)
[d] European Monitoring and Evaluation Programme (Vestreng et al., 2007)
[e] The Big Bend Regional Aerosol and Visibility Observational Study (Kuhns et al., 2005)
[f] Criteria Air Contaminants Van Donkelaar et al. (2012)
[g] National Emissions Inventory (http://www.epa.gov/ttnchie1/net/2005inventory.html)
[h] Li et al. (2017)
[i] The Quick Fire Emissions Dataset (Darmenov and da Silva, 2015)
[j] Yevich and Logan (2003)
[k] Takahashi et al. (2009)
[l] The Simple Biosphere (Messerschmidt et al., 2012)
[m] Baker et al. (2006)
[n] International Comprehensive Ocean–Atmosphere Data Set (Lee et al., 2011)
[o] Aviation Emissions Inventory Code (Stettler et al., 2011)
[p] Fung et al. (1991)
[q] Bloom et al. (2017)
[r] The production of CO from NMVOCs and $CH_4$ is calculated with the GEOS-Chem full chemistry simulation from simulated monthly CO chemical production rates using biogenic NMVOC emissions from the Model of Emissions of Gases and Aerosols from Nature (MEGAN) (Guenther et al., 2012), anthropogenic NMVOC emissions from the Reanalysis of the Troposhperic chemical composition (RETRO) inventory (Bolshcer et al., 2007) and biomass burning NMVOC emissions from GFEDv3 (Fisher et al., 2017).
[s] Fisher et al. (2017)
[t] The chemical production of $CO_2$ is calculated based on monthly CO loss rates from the GEOS-Chem full chemistry simulation
[u] Nassar et al. (2010)
[v] Park et al. (2004)





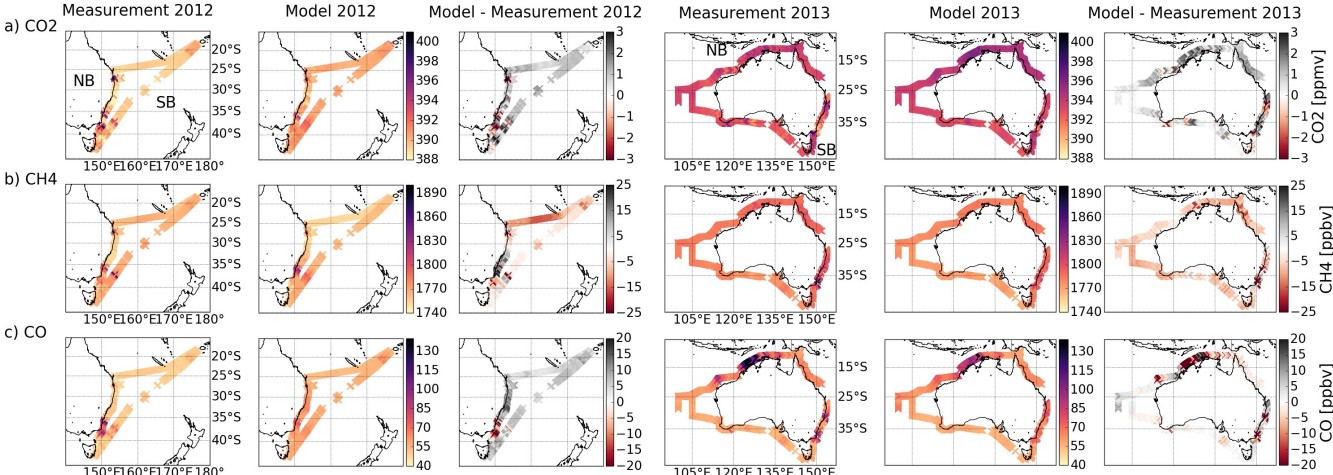

**Figure 3.** Measured and modelled $CO_2$ (a) $CH_4$ (b) and CO (c) concentrations from the ship cruises in 2012 (left) and 2013 (right). The model-measurement difference is also shown for each gas and ship cruise.

## 4 Observed and modelled $CO_2$, $CH_4$ and CO distribution around Australia

Figure 3 shows the measured and modelled $CO_2$, $CH_4$ and CO, and the difference between measurements and model, in 2012 and 2013. In both years, the three gases show similar spatial distributions, indicating their likely co-emission.

In 2012 we observed high concentrations with repeated co-enhancements of all three gases detected along the east coast
(NB part) at 27°, 28°, 32°, and 35° - 38°S, and co-enhancements of only $CO_2$ and CO at 38°S. These are all near urban and industrial areas, indicating the anthropogenic influence at these hotspots. For both years the most enhancements were observed along the east coast, but relative to the 2012 measurements, the 2013 enhancements were dominated by co-enhancements of only two gases, $CH_4$ and CO, and with more pronounced individual enhancements. The ship track was the same along the east coast in both years; however, most of the enhancements observed in that region differed. These results suggest that the
5 different time period of the measurement collection (April/May 2012 compared to September 2013) and transport patterns could have affected the difference in the spatial distribution of these gases. Reanalysis data from GEOS-Chem MERRA2 meteorology show weak easterly winds along the east coast (30 - 34°S) during the 2012 cruise compared to stronger westerly winds during the 2013 cruise (Supplement, Figure S2). The stronger 2013 winds may explain the more well-mixed nature of the enhancements relative to the more distinct enhancements observed in 2012.
The model reproduced most of the observed enhancements along the east coast in 2012, but not in 2013 (Figure 3). The model also showed additional enhancements in 2013 that were not seen in the measurements, such as the high $CO_2$ values at 35°S (SB part). To understand the drivers of the observed enhancements and the difference between the modelled and measured enhancements, we use modelled tracers from the GEOS-Chem model (Sect. 3). Figure 4 shows the latitudinal enhancement of





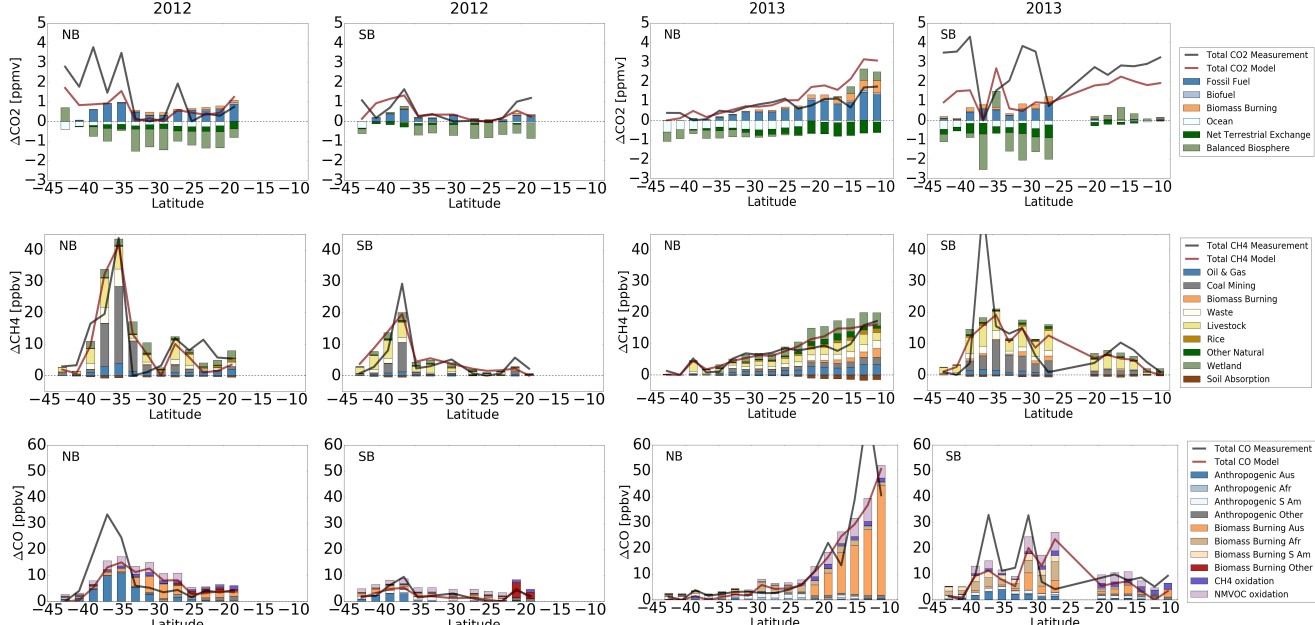

**Figure 4.** Measured (black) and modelled (red) $CO_2$, $CH_4$ and CO latitudinal enhancements (lines) and modelled source contributions (stacked bars) in 2012 (left) and 2013 (right) for the Northbound (NB) and Southbound (SB) sections of the ship cruises. All the data were averaged per $2°$ latitude bands after the measurement-model averaging. The enhancements were calculated based on the difference between the individual $2°$ latitudinal values and the minimum value during each section.

the measured (black) and modelled (red) concentrations, with different modelled tracers (stacked bars) that represent sources

and sinks averaged for every $2°$ latitude after the measurement-model averaging. The latitudinal enhancements were calculated based on the difference between the individual $2°$ latitudinal values and the minimum value of each gas during the section in question (e.g., 2012 NB). With this calculation the contribution of each gas and tracers are treated independently between sections since the change of the gases are calculated relative to the section in question only. If not stated otherwise, the enhancements refer to these latitudinal enhancements that include both the broad scale change of each gas with latitude and the enhancements due to different local or regional sources.

As shown in Figure 4, the model primarily attributes the 2012 east coast enhancements (25 - 44°S) to anthropogenic sources, including fossil fuel for $CO_2$, coal, livestock, oil, gas and waste for $CH_4$, and fossil and biofuel for CO. A previous study by

5 Buchholz et al. (2016) also showed that anthropogenic sources have a strong impact on measurements collected on the east coast. Southern hemisphere biomass burning is more pronounced in September (2013 SB) than April/May (2012 NB) and the model does show a larger influence from biomass burning from Australia and transported from other regions (Africa, South America) for all three gases along the east coast in 2013.



In 2012, enhancements of all three gases are also observed away from the coast on the SB part near Fiji (21°S) and around 38°S, 153°E on the way from Fiji to Hobart. Co-enhancements of only $CH_4$ and CO are observed on the same measurement path around 41°S, 150°E off the north-east coast of Tasmania. The modelled tagged tracers indicate that the high measured concentrations near Fiji arise from a combination of transported biomass burning and anthropogenic sources. The wind patterns and the modelled sources show that the high concentrations observed at 38°S, 153°E, downwind from the southeast Australian coast, are transported anthropogenic sources for all three gases. The model tracers show the same source influences in this downwind region as those observed nearer to the east coast; however, the model underestimates the strength of the transported enhancements due to either underestimated emissions or the influence of numerical diffusion on transport. The SB voyage occurred several weeks after the NB voyage up the east coast, so enhancements with similar source profiles do not necessarily indicate the same enhancement events. For $CH_4$, the transported amounts observed in the downwind region were higher than those observed along the coast during the NB leg (Figure 3), indicating that even if these enhancements derive from the same urban source, the source was stronger during the later (SB) trip than during the earlier (NB) trip. Based on the transport patterns and modelled sources the enhancements at 41°S, 150°E are due to transport from the northeast coast of Tasmania. The main source driving the observed $CH_4$ enhancement along the Tasmanian coast is emission from livestock, in contrast to the strong coal burning emissions observed along the southeast mainland coast.

No significant enhancements were observed along the south and west coasts (2013 NB, 45°S to 25°S); however, there is a gradual increase of all three gases towards the tropics. The model indicates that for $CO_2$ the increase is driven by fossil fuel emissions, biomass burning, changes in the biosphere and also a decrease of the ocean sink, which together result in higher $CO_2$ in the northern parts of Australia. For CO, the latitudinal increase is mainly due to increased biomass burning and NMVOC oxidation, while for $CH_4$, both anthropogenic and natural sources showed a gradual increase with latitude. We attribute a significant part of the $CO_2$ fossil fuel and anthropogenic $CH_4$ sources in the northern parts of Australia to transport from the northern hemisphere due to this gradual increase and the diffused enhancements. Based on the regionally tagged CO tracers the largest contribution to the anthropogenic sources in the northern parts is attributed to transport from regions as Asia, Indonesia and elsewhere in the northern hemisphere (Figure 4, NB section, 2013).

The measurements along the northwest and northern coasts were taken in July/August (NB 2013), when the ITCZ is situated to the north of Australia (Supplement, Figure S3), and Australia is chemically isolated from the northern hemisphere. For long-lived gases like $CO_2$ and $CH_4$, we expect interhemispheric transport to induce a latitudinal gradient throughout the year. Our modelled tracers did show contribution from sources transported from the northern hemisphere but the impact of this transport is expected to be small during austral winter when our measurements were taken. During this period a minimum in $CO_2$ and $CH_4$ is observed in the northern hemisphere due to (boreal) photosynthetic uptake and OH loss, in contrast to corresponding maxima in the southern hemisphere. Based on the measurements along the southwest, west and northwest coasts, we observe a background latitudinal gradient with a standard error of $0.019\pm0.003$ ppm deg$^{-1}$ for $CO_2$, $0.34\pm0.02$ ppb deg$^{-1}$ for $CH_4$ and $0.82\pm0.05$ ppb deg$^{-1}$ for CO. The model showed a stronger latitudinal gradient $0.098\pm0.005$ ppm deg$^{-1}$ for $CO_2$, $0.61\pm0.02$ ppb deg$^{-1}$ for $CH_4$ and $1.09\pm0.07$ ppb deg$^{-1}$ for CO.



In the northern tropical region we observe enhancements and a rise of all three gases between 12° and 20°S (2013 NB). This is likely to arise from biomass burning that occurs during the late dry season (August-September), which is characterised by frequent wildfires (Edwards et al., 2006a). The model captured this rise, but did not fully reproduce the strength of the enhancements. For all three gases it underestimated the source from biomass burning, it did however overestimate a CO enhancement around 12°. Based on the modelled CO tracers (Figure 4), the biomass burning enhancements along the north coast (NB, SB 2013; 10 - 25°S) mainly originated from Australia. Transported biomass burning from Africa was present along the the west coast (NB 2013; 25 - 35°S), while the east coast (SB 2013; 25 - 45°S) was affected by biomass burning from both Africa and South America.

To examine the transported amounts from fires we used data from the MODIS (Moderate Resolution Imaging Spectroradiometer) instrument, and global winds from the MERRA2 reanalysis. Figure S4 (Supplement) shows the total fire pixels from MODIS detected between three weeks and one week prior to each of the seven ship cruises segments in 2012 and 2013, along with monthly mean wind fields. The figure suggests that South American fires prior to the 2013 SB transit along the east coast (September 2013) were stronger than before the 2013 NB transit along the west coast (July 2013). This explains the greater South American biomass burning influence along the east coast relative to the west coast. Strong fires were also observed in Africa prior to both the NB and SB transits in 2013. However, the fires before the SB transit were more spread out along the east and south areas of Africa, and more coincident with the westerly winds, relative to the fires observed during the NB transit. This resulted in more biomass burning emissions transport to the Australian east coast during September and less to the west coast in July.

For both years, we identified sections where no enhancements were observed and used these to quantify background amounts for the gases. During 2012, all three gases were the least variable during the NB section from Brisbane to Fiji in the Coral Sea and on the SB section between 155° and 173°E in the Tasman Sea. During 2013 no enhancements were observed sailing west in the NB section over the Indian Ocean. The locations of these regions are shown in Figure 5 (top panel). The background section mean mole fractions of the gases, both measured and modelled, are shown in Figure 5 and in Table S2 (Supplement). The measurements in the three regions are consistent with the expected temporal and latitudinal variations of these gases. The amounts of all three gases were higher in the Indian Ocean than in the two other regions, due to the interannual and seasonal variability between the periods when the measurements were collected (July 2013 compared to May-June 2012). The amount of $CH_4$ and CO was higher in the Tasman Sea (June 2012) relative to the Coral Sea (May 2012), presumably due to the one-month difference in the measurement timing. $CO_2$ showed minimal difference between the Tasman Sea and Coral Sea background regions, but with lower values in the Tasman Sea, presumably due to the weaker oceanic sink closer to the tropics (Takahashi et al., 2009). The model overestimated the background values for $CO_2$ and CO and underestimated the background $CH_4$ in all three regions. The model-measurement residuals were consistent for each gas in all three background regions showing that the sources or sinks acting on a broader scale need further constraints.

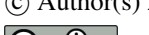



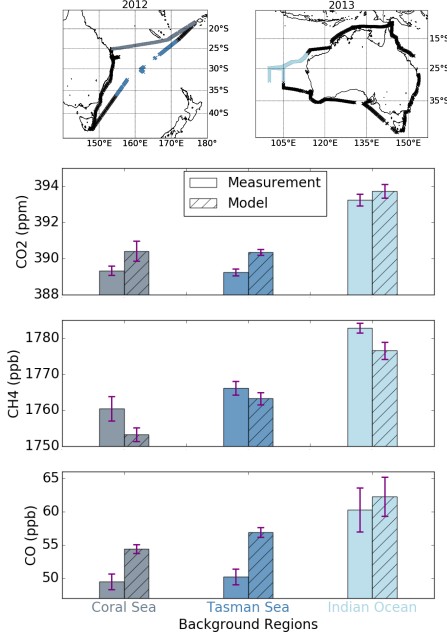

**Figure 5.** Measured and modelled $CO_2$, $CH_4$ and CO concentrations from the ship cruises in different background regions with one standard deviation. The location of the sections where background values were observed are shown on the map. The measurements in the Coral Sea (gray) were collected during May 2012, in the Tasman Sea (dark blue) during June 2012 and in the Indian Ocean (light blue) during July 2013.

## 5  Source variability with respect to scale

To assess how much each source and sink contribution varied at short (local) versus long (regional) scales along the four measurement sections (NB and SB, 2012 and 2013), we separated the total amount of each gas into background values (Figure
30  6a) and enhancements (Figure 6b). The bottom plots in Figures 6a and 6b represent the percentage change of each model tracer relative to the tracers during a given measurement section, while the top plots represent the absolute change in a given tracer relative to the first measurement section (2012 NB).

Figure S5 (Supplement) illustrates the process of separating the measured and modelled data into background values and enhancements. We first averaged the data into $0.1°$ latitudinal values (after the measurement-model averaging described in Section 3), and for each section we calculated the change of all three gases from one latitude bin to another. Based on these changes (e.g. $\delta$CO, Figure S5, Supplement) we examined different values to choose a threshold value that most clearly separates the
5  background regions from the enhancements for each section separately. For changes below the threshold value, the measured and modelled points were classified as background regions, and enhancements if the change between the points was above the threshold value. The threshold values for each section can be found in Table S3 (Supplement). The background values



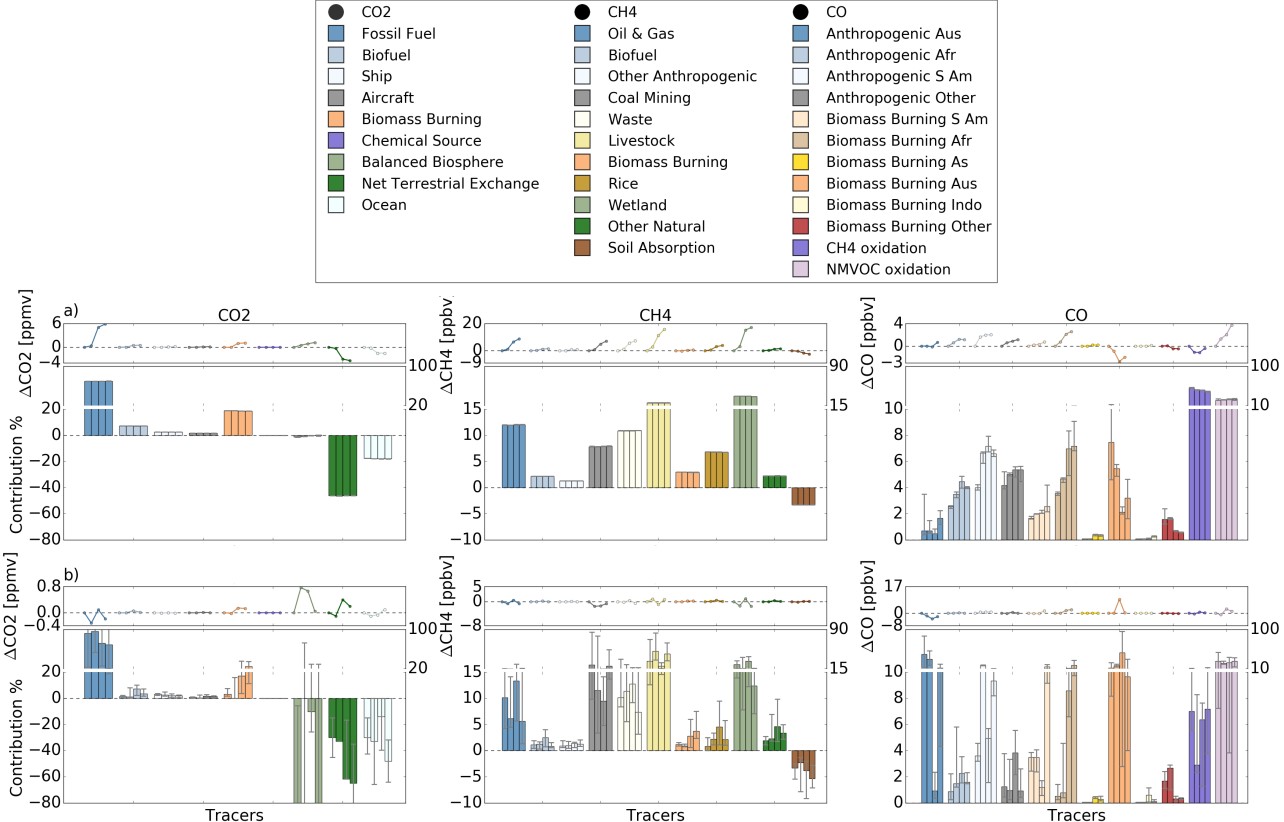

**Figure 6.** $CO_2$, $CH_4$ and CO tracer contribution across the 4 measurement section in 2012 and 2013 (2012 NB, 2012 2B, 2013 NB, 2013 SB, from left to right) for the background (a) and enhancements (b). We separated out the total amount of each gas into background values and enhancements to examine the impact of different time and spatial scale on the change of the sources. The percentage change (bottom plots) show the contribution of each tracer during a specific trip while the concentration change (top plots) is the change of each tracer across the four sections relative to the first section (2012 NB). The contributions are calculated based on the median for each section and the uncertainties represent the 25th (lower error bar) and 75th (upper error bar) percentile.

were additionally filtered to only include data within one standard deviation of the mean. Due to the influence of the latitudinal gradient on the background values, we used a moving mean and standard deviation. Finally, we calculated the relative values of the enhancements based on the difference between the amount of gas at each individual $0.1°$ latitudinal value and the minimum value during the specified sections, as in Section 4. Table S4 (Supplement) provides a statistical comparison of the measured and modelled total, background-only and enhancement-only values.

The source and sink contributions to the background (Figure 6a) values showed the same behaviour as the source and sink contributions to the total amounts (Figure S6 and Table S5, Supplement), but with less variability. Only the CO local sources





(Australian anthropogenic and biomass burning emissions) and African biomass burning showed any difference between background and total amounts. As a result, only the background values are discussed here, but the background analysis also applies to the total amount of each gas.

Our model results suggest that fossil fuels followed by biomass burning contribute the most to background and total $CO_2$ (Figure 6a). Both the biosphere and the ocean were net sinks during all four measurement sections, with a net contribution (-64±0.1%, averaged along the four sections with one standard deviation) about 6% less than the amount of $CO_2$ emitted from fossil fuels alone (69.9±0.2%).

For $CH_4$, wetlands were identified as the biggest background source followed by emissions from livestock, oil, gas and waste. Emissions from coal mining and rice were smaller, but still important. The remaining sources contributed less than 3% each. The $CH_4$ soil absorption tracer represents a sink that is similar in magnitude to the $CH_4$ source from biomass burning, as seen previously by Dalal et al. (2008) for Australia, and their quantification of the contribution of different anthropogenic sources is consistent with our findings here.

For CO, chemical production from $CH_4$ and NMVOCs were the biggest contributors to the background and total amounts (70±2%). This shows that the CO burden in Australia and the southern hemisphere is largely controlled by secondary CO production, consistent with findings from Zeng et al. (2015) that biogenic emissions provide the largest CO background contribution. Biomass burning, both transported and from Australia, is responsible for 14±1% of the total simulated CO, from which 68±12% is attributed to transported biomass burning, with the highest amounts originating from Africa, followed by South America, as seen previously by Gloudemans et al. (2006) and Ridder et al. (2012). Anthropogenic processes contribute 16±2% to the total CO, 90±6% of which is transported (mainly from South America).

In the model, the $CO_2$ and $CH_4$ enhancements (Figure 6b) were generally driven by similar sources to the background amounts (Figure 6a). For $CO_2$, the biospheric influence is more pronounced in the enhancements than in the background. For $CH_4$, anthropogenic sources (especially coal mining) contribute more to the enhancements than to the background, while wetlands (the biggest contribution to the $CH_4$ background) contributes considerably less to the enhancements. Fraser et al. (2011) showed that at a single site on the east coast (Wollongong), coal mining was the largest source of $CH_4$ enhancements above background (60%). Our results suggest that coal mining (21%) and emissions from livestock (28%) are the largest contributors to the enhancements along the east coast in 2012 (leftmost gray bar in Figure 6b).

The CO enhancements were less affected by the tracers that contributed the most to the background, since these tend to be spatially uniform sources. While total and background CO amounts were dominated by secondary sources ($CH_4$ and individual NMVOC oxidation), the enhancements were largely driven by primary CO emissions from biomass burning and anthropogenic sources, with stronger influence from Australian sources than from long-range transport. The CO enhancements also showed significant regional variability.

For all three gases, the enhancements above the background were dominated by temporally and spatially variable sources and sinks, displaying significant variability both within each section and between the four sections. In contrast, the $CO_2$ and $CH_4$ sources and sinks contributing to the background showed minimal variability between the four measurement sections.



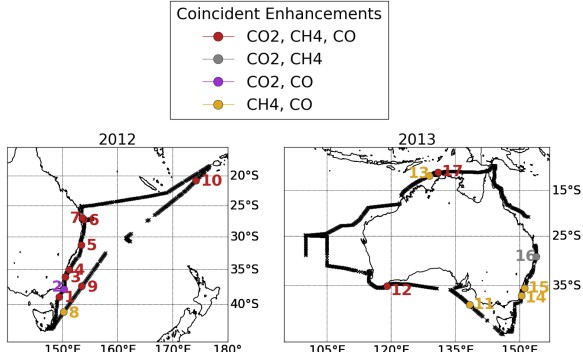

**Figure 7.** Location of the 17 events during which we observed co-enhancements of $CO_2$, $CH_4$ and CO. The red numbers represent events with coincident enhancements in all three species, grey numbers represent co-enhancements in $CO_2$ and $CH_4$ only, purple numbers represent coincident enhancements in $CO_2$ and CO only, and yellow numbers represent coincident enhancements from $CH_4$ and CO only. The black line represents the measurement track during 2012 and 2013.

The CO background sources varied somewhat between the four sections (Figure 6a), due to the shorter CO lifetime, but this background variability was still less than the variability seen in the CO enhancements (Figure 6b).

## 6  $CO_2$, $CH_4$ and CO correlations and co-enhancements

The spatial distributions of the three carbon gases (Figure 3) showed similar enhancement patterns, suggesting that the gases were co-emitted. From these coincident and co-located enhancements, we estimated enhancement ratios (ERs) from both the measured and modelled values, averaged into 0.1° latitudinal bands, across the 4 different sections after the measurement-model averaging. We defined the ER between two species as the slope between the enhancements of the two species calculated using linear regression (Turnbull et al., 2011). Our enhancement ratios also reflect the emission ratios between $CO_2$, $CH_4$ and CO, since these are relatively longed lived gases, and the impact of chemistry on the observed enhancements should be minimal. For the purposes of calculating the ER, the enhancement was defined as the difference between the maximum and minimum value of each gas during the specific co-enhancement event. This definition removes the potential impact introduced by the changing background concentrations between the three gases. Unlike the enhancements discussed earlier, the enhancements used to define the ERs are not affected by latitudinal gradients, and they are not influenced by the changes due to latitudinal or other broad-scale changes.

We use this information to evaluate mismatches between the model and the observations and specifically to determine whether (1) the modelled source profile is correct (i.e., same ERs as in the observations) but with the wrong magnitude for the source or (2) the model has a missing or incorrect source (different ERs). Figures S7 and S8 (Supplement) show species-species linear regressions for events when we observe coincident enhancements in at least two gases.



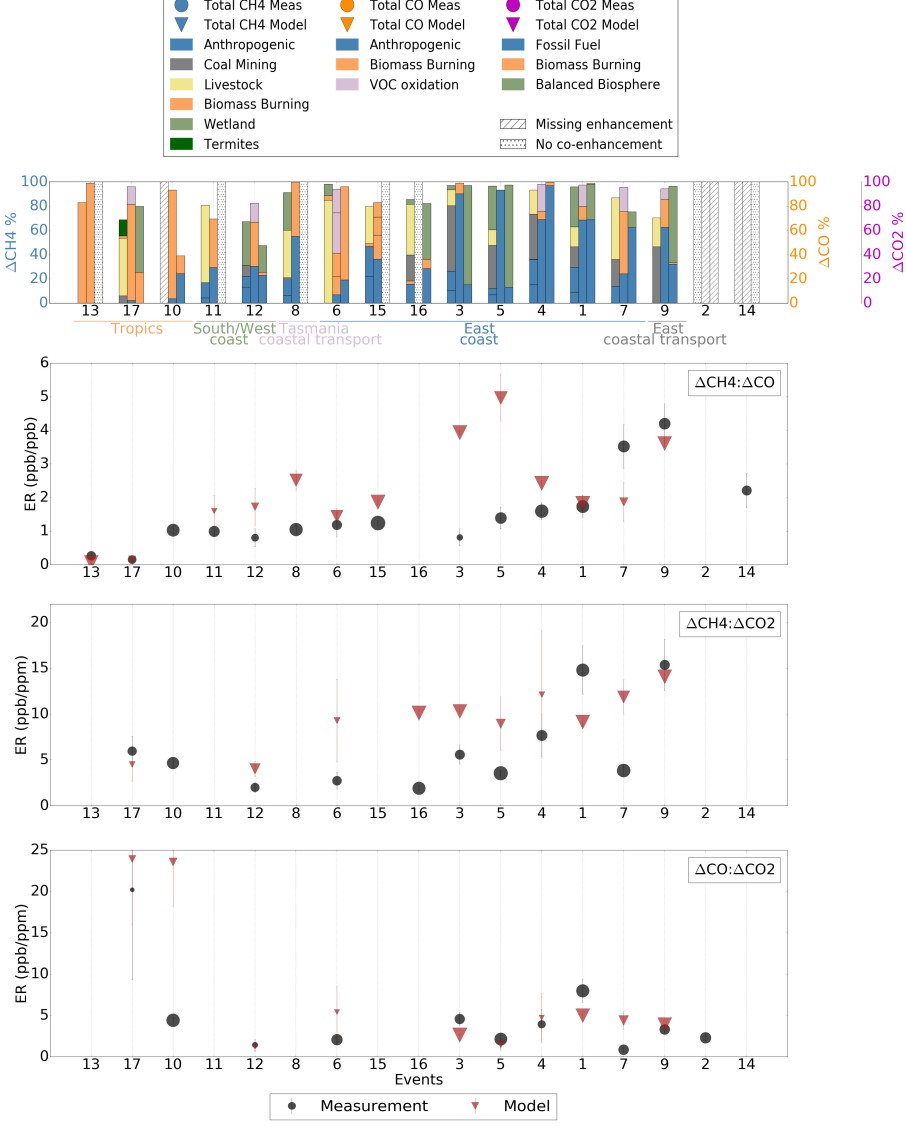

**Figure 8.** The top plot shows the contribution of different tracers (stacked bars) to the modelled enhancements of $CH_4$ (first bar), CO (second bar) and $CO_2$ (third bar) for the 17 events when we observed co-enhancements of the measured gases. The bottom plots show the measured (circle) and modelled (triangle) enhancement ratios for $\Delta CH_4:\Delta CO$, $\Delta CH_4:\Delta CO_2$ and $\Delta CO:\Delta CO_2$, the error bars represent the standard error. The size of the markers represents the correlation coefficient between the species during the coincident enhancements. Enhancement ratios and tracer contributions from the model are only shown for events when the model also saw evidence of co-enhancement. The events are ordered based on both the source type and region where it occurred.



From the measurements, we found evidence of co-enhancements during 17 events. The locations of these events are shown in Figure 7. The ERs and correlation coefficients are also summarized in Table S6 (Supplement). All events except event 3, 4, 12 and 17 showed correlations of $r > 0.80$ between the species during the coincident enhancements. The 2012 measurements generally showed co-enhancements of all three gases while the 2013 data generally showed individual enhancement or enhancements of only two species. Of the 17 events identified in the measurements, the model reproduced co-enhancements for 14 (all except 2, 14, and the $CH_4$ enhancement in 10), however, underestimated the magnitude of most enhancements.

## 6.1 Enhancement ratios and source signatures

Figure 8 (top panel) shows the modelled sources that contributed to co-enhancement events from which we derived enhancement ratios. The measured ERs (bottom panels) are shown as circles, with triangles for the corresponding modelled ERs (only for events when the model simulated similar co-enhancements). The difference between the measured and modelled $CO_2$, $CH_4$ and CO enhancements and ERs during each event is shown in Figure S9 (Supplement).

The modelled tracers suggest there is a relationship between the $\Delta CH_4$:$\Delta CO$ ERs and the sources driving the enhancements. Both the measurements and model showed low ERs for events caused by natural processes (mostly biomass burning, orange), higher ERs for events with mixed natural and anthropogenic signatures, and the highest ERs for events dominated by anthropogenic sources (blue/grey). The balance of sources varies regionally, so the lowest ERs were observed in the tropics due to the impact of stronger natural emissions. Higher ERs were seen along the south and west coasts due to the influence of both natural and anthropogenic sources. We found the highest ERs along the east coast due to the impact of different industrial areas.

The patterns are similar for $\Delta CH_4$:$\Delta CO_2$ ERs, with higher ERs from anthropogenic processes. For $\Delta CO$:$\Delta CO_2$ we found the highest ER for event 17, which is driven by biomass burning, suggesting that biomass burning is the process that produces the most CO relative to $CO_2$ and $CH_4$. The lowest measured $\Delta CO$:$\Delta CO_2$ ERs were identified for events 12 and 7, which derived from anthropogenic sources for both gases combined with additional biomass burning and VOC oxidation for CO and biosphere influence for $CO_2$.

The co-enhancements and events detected along the east coast highlight the anthropogenic influence in this part of Australia. 10 events (events 1, 2, 3, 4, 5, 6, 7, 14, 15, 16) were identified along the east coast, all with dominant anthropogenic signature, and one event (event 9) was detected 400 km off the east coast. The measured and modelled ERs seen during event 9 showed similar values, with the modelled tracers suggesting that this enhancement has an anthropogenic origin, and originates from the east coast due to the similar source composition. The ERs along the east coast were mainly overestimated by the model.

One event (event 8) was observed off the northeast coast of Tasmania. Despite being located in the vicinity of the events observed along the east coast, the $\Delta CH_4$:$\Delta CO$ ER for event 8 is lower than most of the enhancements observed along the east coast. In contrast to the $CH_4$ source contribution along the east coast the main sources are wetlands and livestock, while most of the events along the east coast had coal mining as a dominant source, pointing to a weaker anthropogenic influence from the northeast coast of Tasmania relative to the Australian east coast.





The biggest difference between the measured and modelled ERs when $CH_4$ was co-emitted was during events 3, 4, 5, 7 and 16 (all located along the east coast). The model overestimated the ERs for events 3, 4, 5 and 16 for both $\Delta CH_4:\Delta CO$ and $\Delta CH_4:\Delta CO_2$, while for event 7 it overestimated the $\Delta CH_4:\Delta CO_2$ and underestimated the $\Delta CH_4:\Delta CO$ ER. All the events with the highest modelled ERs (when $CH_4$ is emitted) have coal mining as the dominant source, which suggests that this source was overestimated in the model for events 3, 4, 5, 7 and 16. The fact that the $\Delta CH_4:\Delta CO$ ER during event 7 was underestimated shows that the biomass burning source of CO was too high relative to $CH_4$ and $CO_2$, since the $\Delta CO:\Delta CO_2$ ER was also overestimated by the model.

Prior work on $CH_4$ showed that globally anthropogenic emissions from livestock, landfills and other minor sources are underestimated in EDGARv4.2, coal emissions are overestimated while oil and gas was found to be underestimated globally but overestimated in certain regions (North America, contiguous United States) (Wecht et al., 2014; Turner et al., 2015). Based on co-variations between $CO_2$, $CH_4$ and CO along the east coast we show that the overestimation of the source from coal mining is also present in the emissions from Australia. The only event when coal mining was a dominant source and the model showed similar ER as the measurements was during event 9.

Along the south and west coasts, the sources reflect a mixture of anthropogenic and natural emissions. Relative to the east coast, the ERs were lower for these events (events 11 and 12). The source signatures were similar to some events observed along the east coast with mixed biomass burning and anthropogenic sources, like events 6 and 15. During event 12 the model showed similar ER as the measurements for $\Delta CO:\Delta CO_2$, it overestimated the $\Delta CH_4:\Delta CO_2$ ER, while the $\Delta CH_4:\Delta CO$ ERs were overestimated for both event 11 and 12. This overestimation and the greater difference between the measured and modelled CO enhancements relative to the $CO_2$ and $CH_4$ enhancements (Figure S9, Supplement) suggests that the source from biomass burning was underestimated in the model for both events, since biomass burning was the dominant CO source.

The north coast and tropics were mostly influenced by biomass burning (events 10, 13, 17). The model reproduced the $\Delta CH_4:\Delta CO$ ER during event 13 (when no $CO_2$ enhancement was observed), while for event 17 it reproduced the $\Delta CH_4:\Delta CO$ ER, slightly underestimated the $\Delta CH_4:\Delta CO_2$ ER, and overestimated the $\Delta CO:\Delta CO_2$ ER. These differences were potentially caused by the coarse $2°x2.5°$ resolution of the GEOS-Chem model. With such coarse resolution, the strength of local sources is diffused. The resolution likely affected event 17, when the observed enhancements were weaker and less distinct than those observed during other events. The model overestimated the $\Delta CO:\Delta CO_2$ ER during event 10. Based on the measured-modelled enhancement difference (Figure S9, Supplement) the CO enhancement was overestimated by the model and the modelled $CO_2$ enhancements was underestimated. The difference in the modelled ER is hence likely due to the overestimated strength of the biomass burning source in CO and its underestimation in $CO_2$, since it was shown as a dominant source. The model did not reproduce the $CH_4$ enhancement at all for event 10, pointing to a missing source in the model.

## 6.2 Summary of co-enhancements and implications for missing sources

Using the derived ERs more broadly and linking them to a specific source signature is challenging due to the mixture of sources during the co-enhancement events. From the 17 events only one (event 13) showed contribution from only one source (biomass burning) while all the other co-enhancements were due to a mixture of sources. However, we found these ERs to





be representative in identifying the prevailing processes driving the sources (natural, anthropogenic or mixed), determining sources that are underestimated/overestimated in the model and in identifying the source signatures not captured by the model.

30    Based on our derived ERs we identified the missing sources in the model during events 2, 14 and 10. Events 2 and 14 correspond to a similar region along the east coast, but with one year difference. Event 2 was observed in 2012, and its measured ER was similar to the ER corresponding to event 5. This suggests that this missing source is a combination of anthropogenic (fossil and biofuel) emissions, with an additional natural biosphere source for $CO_2$. The ER for event 14 in 2013 shows a value closest to the modelled ER during event 4, which was observed in the same region in 2012. The modelled sources point mainly to an anthropogenic signature of the missing source for both $CH_4$ (oil, gas, coal mining, livestock) and CO (fossil and biofuel) during this event. The measurements showed enhancements for all three gases during event 10, but the model failed to capture the $CH_4$ enhancement. The sources of $CO_2$ and CO suggest that the missing $CH_4$ source is a combination of biomass burning and anthropogenic sources, with biomass burning being the dominant source, while the similarity between the measured ERs

during events 10, 6 and 11 suggest there is also a significant contribution from livestock.

## 7  Conclusions

We have used in-situ FTIR measurements collected in two consecutive years from a ship that circumnavigated Australia to construct a map of near-surface atmospheric $CO_2$, $CH_4$ and CO distributions around Australia. Using tagged simulations from the GEOS-Chem model, we estimated the contribution of different sources to the total and background amounts of each gas

and identified the drivers of their short-term enhancements. Co-variations between the different measured and modelled gases were used to identify common sources of all three carbon greenhouse gases and to understand the origin of the differences between measured and modelled quantities.

We found significant regional variability in the dominant source contributions along the Australian coast. The Australian east coast was dominated by anthropogenic sources, the south and west coasts showed a mixture of anthropogenic sources and

biomass burning, and the north coast was influenced primarily by natural sources (biomass burning) for CO, anthropogenic (fossil fuel) for $CO_2$ and a a mixture of anthropogenic and natural sources for $CH_4$. Relative to the eastern and northern coasts of Australia, measurements along the south and west coasts showed the least variability. We used these regions to quantify latitudinal gradients for $CO_2$, $CH_4$ and CO. Based on the measurements in the Coral and Tasman Seas in 2012 and the Indian Ocean in 2013 where the air was relatively clean and unaffected by anthropogenic sources transported from the coast, we

estimated the background levels of all three gases. Background concentrations were lowest in the Coral and Tasman Seas, consistent with expected growth in carbon gases from 2012 to 2013.

Our model results suggest that fossil fuels (69.9±0.2%) followed by biomass burning (18.7±0.1%) contributed the most to total $CO_2$ and its background values. For $CH_4$, wetlands (33.1±0.1%) were identified as the largest background source, followed by emissions from livestock (20.59±0.05%), oil and gas (12.01±0.03%) and waste (10.90±0.01%). For CO, sec-

ondary chemical production from $CH_4$ and NMVOCs was the biggest contributors to the background (70±2%). Episodic enhancements in $CO_2$ and $CH_4$ were largely driven by similar sources to the background amounts, although for $CH_4$, the





anthropogenic sources more strongly influenced the enhancements than the background. The CO enhancements were driven by primary CO emissions from biomass burning and anthropogenic sources, with stronger influence from Australian sources than from transported sources.

While the short-term enhancements were driven by local sources, overall we found that sources transported from other regions greatly affect the total amounts of these gases in Australia. For CO, $68\pm12\%$ of the total biomass burning contribution is attributed to transported amounts, mainly from Africa and South America, and $90\pm6\%$ of the total anthropogenic contribution is from transported amounts, with the greatest contribution from South America. Transport from the northern hemisphere was observed closer to the tropics from regions including Asia, Indonesia and elsewhere in the northern hemisphere.

    We observed similar enhancement patterns for $CO_2$, $CH_4$, and CO along the measurement path, pointing to coincident enhancements of these gases. Based on these coincident enhancements, we derived enhancement ratios (ERs) for 17 events. We

found the most events along the east coast, followed by the tropical north coast. The $\Delta CH_4$:$\Delta CO_2$ ERs showed a dependence on both source type and region. We found low ERs for events caused by natural processes, such as biomass burning (tropics and northern Australia), higher ERs for events with mixed natural and anthropogenic sources (south and west coasts) and the highest ERs for events dominated by anthropogenic sources (east coast). The $\Delta CH_4$:$\Delta CO$ ERs also showed higher values for the enhancements that mainly originated from anthropogenic processes. For $\Delta CO$:$\Delta CO_2$ we found the highest ERs for events

driven by biomass burning and the lowest ERs for events that derived from a combination of anthropogenic sources for both gases along with biomass burning and VOC oxidation for CO and biosphere influence for $CO_2$.

    Assumptions in the simulations, lack of time specific emissions and the influence of numerical diffusion on the transport can all introduce uncertainties in the modelled results. Our model results captured the distribution of the measured amounts and the main sources driving the changes of all three gases, but some discrepancies remain. Based on the measured and modelled

ERs, we identified the source signature of the events that were not reproduced by the model. We found coal burning to be over-estimated for $CH_4$ and biomass burning generally underestimated for all three gases, although with CO overestimates during some events. We attribute the missing sources during events that were not reproduced by the model to mainly anthropogenic sources for CO and $CO_2$, oil, gas, coal and livestock for $CH_4$. The exception is along the tropical north coast, where biomass burning is the main underestimated source for all three gases.

Processes driving carbon greenhouse gas changes in Australia were proven to have a large impact on the global carbon cycle and our climate, hence constraints on these processes are essential for predicting future climate change scenarios. Our results show that focusing on simultaneous measurements rather than only one species provides useful additional information in estimating source profiles and contributions. We have shown that the co-variation of $CO_2$, $CH_4$ and CO can be used to constrain the sources of the individual gases, as well identify the drivers of the enhancements that are not reproduced by

models.




## 8   Data availability

All GEOS-Chem model output is available from the authors upon request. GEOS-Chem in an open-source model and the code
is publicly available (wiki.seas.harvard.edu/geos-chem/index.php/Downloading_GEOS-Chem_source_code_and_data). Ship
data was provided by DK and the data will be published in Earth System Science Data and publicly available in Pangaea. The
MODIS data were downloaded form https://neo.sci.gsfc.nasa.gov/view.php?datasetId=MOD14A1_M_FIRE&year=2017.

*Author contributions.*   BB ran the GEOS-Chem simulations, wrote the codes for all the calculations, performed the analysis, and led the
writing of the paper under the supervision and guidance of NMD and JAF. The ship measurements were provided by DK and DWTG who
prepared the FTIR analyser. The observational data set was analysed by DK, DWTG and CM. All authors contributed to editing and revising
the manuscript.

*Competing interests.*   The authors declare that they have no conflict of interest.

*Acknowledgements.*   We acknowledge Australian Research Council (ARC) funding in the form of Discovery Project DP110101948 and
5 Marine National Facility (MNF) transit voyages grants for ship time on three voyages for "Transect Measurements of Greenhouse Gases
in the Marine Atmosphere". The authors are grateful to the CSIRO-MNF technical team for the successful realisation of the measurement
setup and their great support during the voyages. We appreciate the good cooperation with the P&O crew and their helpful hands. We are
also grateful to Graham Kettlewell for the help with the instrument installation aboard the ship. We acknowledge the NOAA ESRL Global
Monitoring Division, Boulder, Colorado, USA (http://esrl.noaa.gov/gmd/) for providing the different background site data. We are also
grateful for the MODIS mission scientists and associated NASA personnel for the production of their data used in this research, obtained
from NASA Earth Observations (NEO) that is part of the EOS Project Science Office at the NASA Goddard Space Flight Center. This
research was undertaken with the assistance of resources provided at the NCI National Facility systems at the Australian National University
through the National Computational Merit Allocation Scheme supported by the Australian Government. The PhD position is supported
by a Discovery Early Career Researcher (DECRA) University Postgraduate Award from the University of Wollongong and the research
undertaken during the PhD is supported by the ARC Grants DE140100178 and DP160101598.



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
