# Peer review of "Simultaneous shipborne measurements of CO2, CH4 and CO and their application to improving greenhouse gas flux estimates in Australia"

_Atmospheric Chemistry and Physics, 2018_

## Referee Comment (RC1) · Anonymous Referee #1 · 10 Feb 2019

This paper discusses analysis of CO2, CH4 and CO measurements around the continent of Australia. They have employed the GEOS-Chem atmospheric chemistry-transport model for simulating the species concentrations, and also different sectors leading up to the total molecular abandances. The manuscript is pretty well writen although a bit descriptive. At times that made it difficult to identify the highligh of a section or a figure. I recommend the authors to find ways to make smaller figures and reduce the length of the text for better communicating the outcomes of this research. For example you could show only the important new results in the figures. Otherwise I only have minor suggestions and comments on the content of the manuscript. The manuscript can be published in Atmos Chem and Phys after the revisions by the au-

thors.

Minor comments: 1. The introduction section discusses at a great length on the importance of Australian natural CO2, but lees is discued in the resuts and discussions, which is mostly about CH4 and CO. A possible place to concretise your text. Similar the Abstract can be shortened.

Figure 3: May be you do not need the Column titled "Model 201x" for both the 2012 and 2013. This would improve clarify and brevity.

Page 11, line 31-32: Can you not use the biomass burning data for the time of your cruises to more accurately attribute the observed enhancements?

Page 11, line 7-8: Can you not put the emissions resulting from the fire pixels in your model, e.g., from GFED, GFAS, FINN etc.?

Figure 5 and associated discussions: I have doubt whether you can treat the Tasman sea as a background region. We see a lot of pollution events at the Cape Grim site when continuous measurements are analysed using global model.

However, the definition to background region may hold good if you employ a high resolution transport model, say at resolution of 10 km!

FIgure 6: Nice plot but difficult to follow, may be consider merging a few sectors in to bigger categories, eg., for CO2 ship and aircraft and chemical sources grouped in to one. Similarly, I see 3 small sources for CO.

The lines in the upper panels can be made more prominent

Page 14, line 8ff: Is this the global scenario? it would interesting know the australian case here! Of if you have discussed the australian case elsewhere, you may not need this here.

Page 14, line 18-19 : Is this because you have wider data coverage or something else, any speculation would be useful here.

Figure 7: Could move the legend to the same row and increase clarity of the data presentation.

page 18, line 1-6: The North America is a bit out of context, this paragraph is not so much needed, except for the fact that your data agree well with the emission inventory of coal mining! I also have a feeling that the ERs are difficult to define "precisely" from atmospheric measurements of atmospheric species of different lifetimes. At least many more events at a specific location is needed for statistically significantly determine the ERs.

---

## Referee Comment (RC2) · Anonymous Referee #2 · 25 Feb 2019

Bukosa et al. present a well thought out modelling analysis of a ship-based greenhouse gas measurement campaign around Australia. Tagged tracer model simulations and ratios between anomalies in CO2, CH4, and CO are used effectively to explore how different source processes impact atmospheric greenhouse gas distributions around Australia and identify biases in the flux estimates. While the work is generally sound, I have some recommendations that I feel will strengthen and clarify the paper.

Framing the analysis: The bulk of the paper is very descriptive, and much of the text works through the data explaining the processes driving each individual ship transect. It is hard work for the reader to find they key scientific findings of the study, and I'm

concerned the paper will not attract the readership it deserves. The authors need to identify a few key results they wish to highlight, and re-focus their discussion of the data on what they need to tell that story.

One approach would be to re-write section 4, so that instead of the analysis going through cruise by cruise, the description focuses instead on what is learned about the key source/sink processes (e.g. fire, fossil, wetlands, etc.) These processes could then have their own sub-sections, to help guide the reader. Then, I would recommend better highlighting the importance and power of the co-enhancements explored in section 6 in the abstract, introduction, and conclusions, to bring this interesting analysis forward a bit. Please take this only as an example of how I might approach the problem, as there are a few equally valid paths the authors could take to focus the paper.

Incomplete use of data: I was surprised that the authors made no use of any of the land-based greenhouse gas measurements available during this time through CSIRO's observing network or TCCON. It is reasonable for the paper to focus on what is learned from the new ship data, but the other long-term data could be a powerful point of comparison.

Emissions fields: The authors have chosen emission fields that, in many cases, do not extend through the time period they wish to simulate (2012-2013), even when more updated fields are available. For example, why use EDGAR 4.2, which ends in 2008, when EDGAR 4.3 continues to 2012 and has been available since 2017? Why use net terrestrial exchanges from TansCom circa 2006, when biosphere estimates from a variety of land models and inverse models are available publicly? I don't think this is likely to cause a fundamental change in the findings, but it is always best practice to use up to date information where possible.

Publication of the data: This manuscript is built upon a measurements campaign, which the authors say will be described in a forthcoming paper by Kubistin et al. I have a high degree of confidence in the capabilities of the measurement team for this work. However, there is a risk in accepting a paper that is underpinned by another manuscript, before that manuscript is fully peer reviewed or accepted.

Minor comments:

Introduction: For multi-species studies such as this one, I think it is very helpful to show a table of source/sink processes, highlighting which gases are affected by the source/sink. This way, the reader sees immediately what the common threads are, even if they don't know a specific gas particularly well.

Introduction: There is a strong focus in the introduction on the importance of Australian fluxes in driving global inter-annual variability in greenhouse gases. However, this paper doesn't shed a lot of light on inter-annual variability. Instead, it is a strong example of how in-depth analysis across multiple trace gases can be a powerful tool in source/sink attribution. I recommend re-framing the introduction this way, to attract the readers that are most likely to be enthusiastic about this work.

Page 5, line 10. The introduction spent so much time talking about inter-annual variability in Australian fluxes, it seems strange to read here that you think this isn't going to cause much error to use repeating fluxes in your model simulations.

In the introduction, the authors highlight the strong variability in greenhouse gas fluxes from Australia. It feels a little strange to read here that the variability to

Figures 6 and 8 are the most exciting results figures of the paper, yet they are quite small and difficult to read. Suggest sizing them up.

There were a few ?? left over from the LaTex formatting. Be sure to check and resolve these.

---

## Author Comment (AC1) · 17 Apr 2019

**Authors Response Reviewer 1 - Simultaneous shipborne measurements of $CO_2$, $CH_4$ and CO and their application to improving greenhouse gas flux estimates in Australia**

Beata Bukosa[1], Nicholas M. Deustcher[1], Jenny A. Fisher[1], Dagmar Kubistin[1,2], Clare Paton-Walsh[1], David W.T. Griffith[1]

[1] School of Earth, Atmospheric, and Life Sciences, University of Wollongong, NSW, Australia

[2] German Meteorological Service, Meteorological Observatory Hohenpeissenberg, Hohenpeissenberg, Germany

*Correspondence to:* Beata Bukosa (bb907@uowmail.edu.au)

We would like to thank the reviewer for the comments and suggestions. Below we have provided a response to each comment along with any subsequent changes to the manuscript. The original review is in bold and our responses are in normal font. Please, find the revised manuscript at the end of this document.

**General Comment**

1. **I recommend the authors to find ways to make smaller figures and reduce the length of the text for better communicating the outcomes of this research. For example you could show only the important new results in the figures.**

We have modified the size of some of the figures to improve clarity (Figure 7, Page 15 and Figure 8, Page 16), and we only present the important new results on Figure 6 (Page 13). We have also reduced the text length, both in the introduction and results sections. All the differences are attached below.

**Minor Comments**

2. **The introduction section discusses at a great length on the importance of Australian natural CO2, but lees is discused in the resuts and discussions, which is mostly about CH4 and CO. A possible place to concretise your text. Similar the Abstract can be shortened.**

We appreciate the suggestion and have modified and shortened both the Abstract and Introduction (Page 1, 2 and 3) to better match the focus of the results.

3. **Figure 3: May be you do not need the Column titled "Model 201x" for both the 2012 and 2013. This would improve clarify and brevity.**

We appreciate the comment; however, we prefer to keep the columns with the modelled data, because we believe they help highlight the instances where the model captures the individual enhancements (not that obvious from the plots showing the model-measurement difference).

4. **Page 11, line 31-32: Can you not use the biomass burning data for the time of your cruises to more accurately attribute the observed enhancements?**

We are indeed using biomass burning emissions (QFED) that match the timing of all the ship cruises (Table 1, Page 4) to identify the sources that drive the observed enhancements. We have clarified this in the main text by adding the following: "These biases suggest that, despite using the year-specific biomass

burning emissions from the QFED inventory, there are still uncertainties in biomass burning emissions that affect simulation of carbon gases." (Page 10, Line 17-18).

5. **Page 11, line 7-8: Can you not put the emissions resulting from the fire pixels in your model, e.g., from GFED, GFAS, FINN etc.?**

The QFED biomass burning emission inventory used here is already based on satellite-detected fire pixels (similar to GFED, GFAS, and FINN, `http://wiki.seas.harvard.edu/geos-chem/index.php/QFED_biomass_burning_emissions`) (Table 1, Page 4). We have clarified this in the main text by adding the following: "This explains the greater South American biomass burning influence along the east coast relative to the west coast, observed by the model using QFED biomass burning emissions, that are based on products from MODIS." (Page 10, Line 26-28).

6. **Figure 5 and associated discussions: I have doubt whether you can treat the Tasman sea as a background region. We see a lot of pollution events at the Cape Grim site when continuous measurements are analysed using global model. However, the definition to background region may hold good if you employ a high resolution transport model, say at resolution of 10 km!**

Although pollution events were observed at Cape Grim, those events were primarily driven by pollution from either inland Tasmania or Victoria; however, our Tasman Sea background region ends about 700 km away from Cape Grim (Figure 5, top plot, Page 12) and it is less likely to be affected by the pollution plumes observed at Cape Grim. In any case, our background definition is based objectively on where the measurements (not the model) showed low variability, indicative of a lack of pollution events. From the measurements, $CO_2$ and $CH_4$ showed the lowest variability in the Tasman Sea relative to the other two background regions (Coral Sea and Indian Ocean), while for CO it showed lower variability than the Indian Ocean and similar variability as in the Coral Sea. We used this range of variability (standard deviation, Figure 5) to identify part of the Tasman Sea as a background region.

7. **FIgure 6: Nice plot but difficult to follow, may be consider merging a few sectors in to bigger categories, eg., for CO2 ship and aircraft and chemical sources grouped in to one. Similarly, I see 3 small sources for CO. The lines in the upper panels can be made more prominent.**

We have now merged a few sectors into groups ($CO_2$ ship and aircraft, $CH_4$ other anthropogenic and biofuel, CO biomass burning Asia, biomass burning Indonesia and biomass burning Other). Additionally, we have removed the $CO_2$ chemical source as described at the end of this document under 'Additional Author's Comment'. We have also improved the size of the upper plot lines (Figure 6, Page 13).

8. **Page 14, line 8ff: Is this the global scenario? it would interesting know the australian case here! Of if you have discussed the australian case elsewhere, you may not need this here.**

All the results presented in these paragraphs refer to the Australian case, based on measurement ship tracks, and the Australian results presented here were not discussed in the previous sections. We have now clarified that in the text by adding a sentence: "All the contributions are calculated along the ship

track, hence results discussed here refer to $CO_2$, $CH_4$ and CO in the Australian region only." (Page 11, Line 31-32)

9. **Page 14, line 18-19 : Is this because you have wider data coverage or something else, any speculation would be useful here.**

Yes it is primarily due to the wider data coverage. The referenced paper only focused on one measurement site along the east coast (Wollongong), while our results are based on the ship track that covers a larger part of the Australian east coast. The referenced paper results are also based on measurements for pre-2012 years. We have now clarified this by adding the following: "Although, the different analysis time periods might have influenced these differences, the main reason for the lower coal mining contribution in our results is due to the wider measurement region along the east coast used to quantify these contributions." (Page 14 Line 21-23).

10. **Figure 7: Could move the legend to the same row and increase clarity of the data presentation.**

This figure was planned as a 1 column figure, which is why the label is at the top. We have now adjusted the position and size of the co-enhancement numbers to increase clarity (Figure 7, Page 15).

11. **page 18, line 1-6: The North America is a bit out of context, this paragraph is not so much needed, except for the fact that your data agree well with the emission inventory of coal mining! I also have a feeling that the ERs are difficult to define "precisely" from atmospheric measurements of atmospheric species of different lifetimes. At least many more events at a specific location is needed for statistically significantly determine the ERs.**

Thank you for the comment. We agree that ERs are difficult to define "precisely" from atmospheric measurements and that more ER events, as well ER derived from additional measurement techniques, would improve the statistical significance of our results, which is why we do not draw conclusions on the other less pronounced sources. For coal mining; however, we found a strong and significant pattern between the different co-enhancement events pointing that this source is overestimated. We feel it is important to use our work to evaluate the performance of emission inventories such as EDGAR, especially in Australia where there has been little to no previous validation due to a lack of available data. We refer to results from other regions (e.g., North America) to show that the biases we see in Australia are not unique. In other words, we wanted to highlight specific regions where this was explored previously, and add Australia as an additional region where coal mining is overestimated, since to the best of our knowledge no one has explored this before. We have now modified the paragraph to make it clear why we discussed the different regions (Page 18 Line 1-7).

**Additional Author's Comment**

In addition to all the reviewer's suggestions, we have removed the results for the $CO_2$ chemical source in the paper, since we have discovered an error with the treatment of this source in the GEOS-Chem simulation. Since the chemical source is distributed throughout the troposphere, its impact on our results for surface air is minimal. We have removed this source from Figure 6 and Table 1 in the main text and

Figure S6 and Table S5 in the Supplement. We have also removed this source from the model description (Page 6 line 17). Figure 1, below, shows the surface concentration of the $CO_2$ chemical source relative to a major (fossil fuel) and a minor (biofuel) source, showing that the magnitude of this source type is low relative to other sources at the surface.

[Figure]

Figure 1: Surface concentrations of $CO_2$ from fossil fuel, biofuel emissions and chemical production. Not the different colour scales for the three panels.

[revised manuscript text omitted]

---

## Author Comment (AC2) · 17 Apr 2019

**Authors Response Reviewer 2 - Simultaneous shipborne measurements of $CO_2$, $CH_4$ and CO and their application to improving greenhouse gas flux estimates in Australia**

Beata Bukosa[1], Nicholas M. Deustcher[1], Jenny A. Fisher[1], Dagmar Kubistin[1,2], Clare Paton-Walsh[1], David W.T. Griffith[1]

[1] School of Earth, Atmospheric, and Life Sciences, University of Wollongong, NSW, Australia

[2] German Meteorological Service, Meteorological Observatory Hohenpeissenberg, Hohenpeissenberg, Germany

*Correspondence to:* Beata Bukosa (bb907@uowmail.edu.au)

We would like to thank the reviewer for the comments and suggestions. Below we have provided a response to each comment along with any subsequent changes to the manuscript. The original review is in bold and our responses are in normal font. Please, find the revised manuscript at the end of this document.

**General Comments**

1. **Framing the analysis: The bulk of the paper is very descriptive, and much of the text works through the data explaining the processes driving each individual ship transect. It is hard work for the reader to find they key scientific findings of the study, and Im concerned the paper will not attract the readership it deserves. The authors need to identify a few key results they wish to highlight, and re-focus their discussion of the data on what they need to tell that story. One approach would be to re-write section 4, so that instead of the analysis going through cruise by cruise, the description focuses instead on what is learned about the key source/sink processes (e.g. fire, fossil, wetlands, etc.) These processes could then have their own sub-sections, to help guide the reader. Then, I would recommend better highlighting the importance and power of the co-enhancements explored in section 6 in the abstract, introduction, and conclusions, to bring this interesting analysis forward a bit. Please take this only as an example of how I might approach the problem, as there are a few equally valid paths the authors could take to focus the paper.**

We thank the reviewer for this comment. We agree and have now modified the Abstract, Introduction, Conclusion and parts of Section 6 to highlight the importance of the co-enhancements for source and sink attribution. We have also modified Section 4, shortened the discussion and separated into anthropogenic sources, natural sources and background regions. Please, find the revised manuscript and modifications at the end of this document

2. **Incomplete use of data: I was surprised that the authors made no use of any of the land-based greenhouse gas measurements available during this time through CSIROs observing network or TCCON. It is reasonable for the paper to focus on what is learned from the new ship data, but the other long-term data could be a powerful point of comparison.**

Our original goal was to integrate all the measurements mentioned by the reviewer; however, we made the decision to focus exclusively on the measurements from the ship voyages because (1) the analysis was

becoming too complex and the (already lengthy) manuscript started to lose focus (2) the main two fixed location sites closest to the ship tracks (Darwin and Wollongong) that have both surface and TCCON measurements lacked surface measurements during the time period of the ship cruises, which limited our ability to use the data together to constrain the sources and sinks. However, we do plan to build on these results in future and additionally explore both the surface and column measurements from other sites in Australia, especially the two sites (Wollongong and Darwin) that have both column (TCCON) and surface measurements.

3. **Emissions fields: The authors have chosen emission fields that, in many cases, do not extend through the time period they wish to simulate (2012-2013), even when more updated fields are available. For example, why use EDGAR 4.2, which ends in 2008, when EDGAR 4.3 continues to 2012 and has been available since 2017? Why use net terrestrial exchanges from TansCom circa 2006, when biosphere estimates from a variety of land models and inverse models are available publicly? I dont think this is likely to cause a fundamental change in the findings, but it is always best practice to use up to date information where possible.**

While it would have been ideal to use more up to date information for some emissions, we were limited by which inventories had already been implemented in GEOS-Chem at the time we ran our simulations. Not all emission inventories are inherently compatible with GEOS-Chem, and it is non-trivial to modify the model to use new emission inventories that have not already been tested in the model. We also prioritised using the same inventories for all three species where possible. For example, while EDGAR 4.3 was partially implemented into the $CH_4$ simulation (with caveats, `http://wiki.seas.harvard.edu/geos-chem/index.php/CH4_simulation`) it was not yet implemented in the CO simulation so we used version 4.2.

4. **Publication of the data: This manuscript is built upon a measurements campaign, which the authors say will be described in a forthcoming paper by Kubistin et al. I have a high degree of confidence in the capabilities of the measurement team for this work. However, there is a risk in accepting a paper that is underpinned by another manuscript, before that manuscript is fully peer reviewed or accepted.**

While measurements from the ship voyages will be presented in more detail in the forthcoming paper by Kubistin et al. (2019), we were careful to include essential information in our paper about the specifications of the instrument used for the measurement collection (e.g. repeatability, accuracy), and we discuss both the measurement collection and calibration process (e.g. reference gases in Section 2). Moreover, the instrument used for the measurements is already fully described in Griffith et al. (2012). We feel therefore that our paper stands on its own without the need for the forthcoming ESSD paper by Kubistin et al. (2019). The Kubistin et al. paper will nevertheless complement this paper and is in the final stages of preparation before submission to ESSD in Q2 2019.

**Minor Comments**

5. **Introduction: For multi-species studies such as this one, I think it is very helpful to show a table of source/sink processes, highlighting which gases are affected by the**

source/sink. **This way, the reader sees immediately what the common threads are, even if they dont know a specific gas particularly well**

Agreed, and the purpose of having all the source and sink fields used by the model for all three gases next to each other in Table 1 (Page 4) was exactly for this reason. We have now additionally highlighted this by adding a sentence in the Introduction: "Table 1 highlights the source and sink fields that are common between the three gases. (Page 2 Line 34-35)).

6. **Introduction: There is a strong focus in the introduction on the importance of Australian fluxes in driving global inter-annual variability in greenhouse gases. However, this paper doesnt shed a lot of light on inter-annual variability. Instead, it is a strong example of how in-depth analysis across multiple trace gases can be a powerful tool in source/sink attribution. I recommend re-framing the introduction this way, to attract the readers that are most likely to be enthusiastic about this work.**

We have now re-framed the Introduction by removing the discussion about the interannual variability and focused more on multiple tracer gas studies for source and sink attribution (Page 1, 2 and 3).

7. **Page 5, line 10. The introduction spent so much time talking about inter-annual variability in Australian fluxes, it seems strange to read here that you think this isnt going to cause much error to use repeating fluxes in your model simulations. In the introduction, the authors highlight the strong variability in greenhouse gas fluxes from Australia. It feels a little strange to read here that the variability to**

We have now clarified and expanded this statement with specific examples why we think that for our measurement time period the repeating fluxes will not introduce a large error in the results (Page 6, Line 5-14) even if a significant interannual variability of these gases was observed in the past for the Australian region.

8. **Figures 6 and 8 are the most exciting results figures of the paper, yet they are quite small and difficult to read. Suggest sizing them up.**

We have now modified the figure sizes (Figure 6, Page 13 and Figure 8, Page 16)

9. **There were a few ?? left over from the LaTex formatting. Be sure to check and resolve these.**

These issues are now resolved.

**Additional Author's Comment**

In addition to all the reviewer's suggestions, we have removed the results for the $CO_2$ chemical source in the paper, since we have discovered an error with the treatment of this source in the GEOS-Chem simulation. Since the chemical source is distributed throughout the troposphere, its impact on our results for surface air is minimal. We have removed this source from Figure 6 and Table 1 in the main text and Figure S6 and Table S5 in the Supplement. We have also removed this source from the model description (Page 6 line 17). Figure 1, below, shows the surface concentration of the $CO_2$ chemical source relative to a major (fossil fuel) and a minor (biofuel) source, showing that the magnitude of this source type is low relative to other sources at the surface.

[Figure]

Figure 1: Surface concentrations of $CO_2$ from fossil fuel, biofuel emissions and chemical production. Not the different colour scales for the three panels.

**References**

[revised manuscript text omitted]